# Identification and Spatiotemporal Analysis of Bikesharing-Metro Integration Cycling

Hao Wu [1,2], Yanhui Wang [1,2,*], Yuqing Sun [1,2], Duoduo Yin [1,2,3], Zhanxing Li [1,2] and Xiaoyue Luo [1,2]

1   College of Resources Environment and Tourism, Capital Normal University, Beijing 100048, China;
    2200902181@cnu.edu.cn (H.W.)
2   Key Laboratory of 3-Dimensional Information Acquisition and Application, Ministry of Education,
    Capital Normal University, Beijing 100048, China
3   Inner Mongolia Autonomous Region Water Conservancy Development Center, Hohhot 010010, China
*   Correspondence: yanhuiwang@cnu.edu.cn

**Abstract:** An essential function of dockless bikesharing (DBs) is to serve as a feeder mode to the metro. Optimizing the integration between DBs and the metro is of great significance for improving metro travel efficiency. However, the research on DBs–Metro Integration Cycling (DBsMIC) faces challenges such as insufficient methods for identification and low identification accuracy. In this study, we improve the enhanced two-step floating catchment area and incorporate Bayes' rule to propose a method to identify DBsMIC by considering the parameters of time, distance, environmental competition ratio, and POI service power index. Furthermore, an empirical study is conducted in Shenzhen to verify the higher accuracy of the proposed method. Their spatiotemporal behavior pattern is also explored with the help of the kernel density estimation method. The research results will help managers improve the effective redistribution of bicycles, promote the coupling efficiency between transportation modes, and achieve sustainable development of urban transportation.

**Keywords:** dockless bikesharing; metro integration; behavior identification; spatiotemporal pattern; urban transport

## 1. Introduction

With the background of global warming and increasingly severe energy security, a low-carbon economy, energy saving, and emission reduction have become important issues for the global economy, politics, and diplomacy. Cities are the centers of human economic activity and the main consumers of energy on earth [1]. Carbon dioxide emissions from private cars in urban road traffic are one of the main sources of greenhouse gas in cities [2]. Therefore, building a green transportation network, creating a green low-carbon transportation city, and guiding the public to prioritize green transportation are important pathways to achieving urban "carbon peaking and carbon neutrality" goals and sustainable development. As an emerging low-carbon urban transport method, dockless bikesharing (DBs) has been used by more citizens every day due to its greenness, low-carbon emissions for the environment, flexibility in short-distance travel, and wide coverage density [3]. Compared to traditional docked bikesharing, DBs is not constrained to rental stations. It can be stop-and-go, while docked bikesharing must be borrowed and returned at fixed stations, so DBs has gradually replaced docked bikesharing as the mainstream of the market in China [4]. In addition to being an independent method of travel, providing connection services to metro stations is also an elementary function of DBs [5]. According to the White Paper of 2017 Bikesharing and Urban Development, DBs has become the fourth most popular method of travel after private cars, public transport, and the metro, with nearly 80% of DBs activities near public transport stations [6]. The emergence of the DBs and metro connection approach has effectively solved the problem of the last kilometer and the first kilometer. It also extended the influence of metro stations to a certain extent,

promoting public transport green travel, making some "quasi-metro houses" which were originally too far away from metro stations and had a long walking time upgrade to "metro houses" [7], further enhancing the happiness and sense of gain in residents participating in green transport.

Exploring the temporal and spatial patterns of DBs–Metro Integration Cycling (DBsMIC) can provide insights into the intrinsic mechanism of urban public transport integration travel, optimizing the integration between DBs and the metro, and thus improving metro transfer efficiency and promoting green and healthy travel for residents. Therefore, this area has received increasing attention from scholars [8–10]. However, with the gradual tightening and refinement of urban road management planning and traffic management policies, the delineation of no-parking zones for DBs drives users to park their bicycles in designated areas. This makes it difficult to accurately identify the origin or destination of user cycling by simply relying on parking points, thus making it hard to uncover the real cycling motivation of users. Similarly, if one simply considers DBs activities within a near metro station area as a feeder mode to the metro [11–13], this will lead to many DBs activities that are not related to the metro integration service being included in the analysis of the integration cycling, resulting in a large error between the research results and the actual travel patterns of users. Therefore, accurately and effectively identifying DBsMIC is currently a major challenge and a prerequisite for exploring the temporal and spatial patterns of integration cycling.

In this context, this paper proposes a method for identifying DBsMIC based on an improved Enhanced Two-step Floating Catchment Area (E2SFCA) and Bayes' rule. Taking Shenzhen as a case study, the method will be used to identify the DBsMIC of residents, and its reliability and advancement will be verified by comparing its accuracy with other methods. Finally, based on the identification results, the spatial and temporal characteristics of DBsMIC of Shenzhen residents will be analyzed, providing a reference for relevant transportation planning and management to promote green travel.

The remainder of this paper is organized as follows. Section 2 offers a review of the literature. Section 3 introduces the study area and the dataset. Section 4 describes the research methods. Section 5 presents and discusses the results, and Section 6 concludes the study.

## 2. Related Work

There are two main sources of data for DBs travel studies; the first are questionnaire survey data [14–17]. On-site questionnaire surveys are common, with sites usually selected in large cities where DBs is widely distributed, such as Beijing, Shanghai, Sydney, and Seattle. However, this method is difficult to collect and implement, time-consuming, and expensive. The limited sample size and spatial coverage of the survey data make it difficult to accurately observe spatial and temporal patterns of wide-scale integration cycling. The second sources are traffic big data [18–21], which allows for effective analysis of DBs integration cycling by capturing massive amounts of data and simulating the real traffic environment. As research progresses, scholars have identified important links between DBs and the metro. For example, Schimohr et al. found a strong spatial relationship between DBs and the metro, suggesting that DBs have rapidly emerged as a new feeder mode to the metro for travelers [20]. Gao et al. explored the impact of built environment factors interacting with DBs travel [21]; they found that factors such as weekdays, or metro stations near a company or restaurant significantly enhanced the impact of DBs travel. The results suggest that DBs plays an important role in metro integration. However, few scholars have researched DBsMIC, mainly due to the difficulty of accurately identifying DBsMIC. Unlike DBs, docked bikesharing has the same user ID for both the rental station swipe card system and the metro swipe card system. Bicycle–metro integration cycling orders can be easily filtered out through the user IDs of the public bicycle swipe card data and the metro swipe card data [22]. In contrast, data of DBs orders are scattered across multiple independent data systems and are not connected to the metro data system. Therefore,

identifying DBsMIC orders is a precondition for the study of DBsMIC. In the study of OFO Corporation [6], orders that start or end within 100 m of the entrance and exit of the metro station are defined as "DBsMIC". Li et al., Wu et al., and Guo et al. have also used a similar hypothesis, where a 100 m range is used to set up a parking ring at metro stations to identify DBsMIC orders [23–25]. However, this hypothesis has an obvious flaw: metro stations are often close to shopping malls, schools, parks, residential areas, etc. As a result, DBs orders within 100 m of a metro station are not always destined for or originated from the metro station. Therefore, filtering out the actual DBs orders that are destined for or originate from metro stations is still a problem that needs to be solved.

On the other hand, research on transportation destination inference provides an idea for solving the problem of DBsMIC identification [26,27]. One possible approach is to reframe the problem of identifying DBsMIC as a DBs destination inference task. In this approach, trace data associated with metro stations can be filtered from the destinations, which may help to more accurately identify DBsMIC.

As for DBs destination identification, Li et al. and Ross-Perez et al. have taken the lead in conducting research, respectively implementing inference of DBs travel destinations based on the gravity model and the E2SFCA [28,29]. The gravity model calculates the correlation between two locations or regions [30], but it ignores the scale of supply points and the competition between them. So, Radke et al. proposed a modified algorithm of the gravity model, the Two-step Floating Catchment Area (2SFCA) [31], which is centered on calculating the ratio of supply and demand between two locations, thus deriving the accessibility of each demand point. Luo et al. proposed an E2SFCA as a deformation of 2SFCA. Compared to traditional methods, E2SFCA segments the range-attenuation function of the search radius and gives weight for accumulation using Gaussian equations, thus allowing a good assessment of the attractiveness of different demand points to supply points [32]. Therefore, this paper plans to conduct a study on the inference of DBs travel destinations based on E2SFCA, and then identify the DBsMIC.

At the same time, the spatiotemporal characteristics and spatiotemporal pattern analysis of DBsMIC as the basis of traffic travel analysis research is less relevant. Remaining at the stage where temporal and spatial characteristics are studied independently of each other, methods are mostly described by mathematical and statistical analysis [24,25]. Wu et al. used volume line graphs and histograms to characterize the temporal feature of DBsMIC in terms of cycling distance and volume, portraying the spatial characteristics of DBsMIC through statistical analysis of station cycling volume and station cycling distance. Guo et al. divided the study period into multiple time dimensions to reflect the spatiotemporal rules of residents' integration cycling in the study area in terms of the statistical distribution of access use and egress use volume. In summary, despite the limitations of previous studies in which spatial and temporal characteristics were analyzed independently, the traditional method of spatiotemporal characterization is still essential for basic research. Based on the traditional spatiotemporal feature analysis, it is necessary to further combine the temporal and spatial features to quantitatively analyze the spatiotemporal distribution patterns of DBsMIC.

## 3. Study Area and Data Sources

### 3.1. Study Area

Shenzhen was chosen as the study area for this paper. Since the reform and opening, Shenzhen has experienced rapid urbanization and expansion, becoming one of the most active regions in terms of population movement and economic activity in China and the world. As a coastal supercity in China, Shenzhen has a developed public transport system as shown in Figure 1. By the end of 2021, Shenzhen metro had opened a total of 12 operational routes. With 234 metro stations and 431 km of operational mileage, more than 40% of public transport travels in Shenzhen are made through the metro [33]. However, most of Shenzhen's metro coverage areas are located in the city center or some suburban areas close to the city center, such as southern Baoan, southern Longhua, and Longgang West,

have experienced rapid development over the past decade, with many residents living in these areas and working in the city center. The separation of jobs and housing has led to a weekday commuter tide in the city, and the burden of transport commuting remains huge. The emergence of DBs has greatly alleviated this phenomenon and effectively improved the operational efficiency of the metro. Up until early 2021, there were nearly 389,000 DBs operating in Shenzhen, covering a wide area. This makes Shenzhen a good case area for a study of DBsMIC.

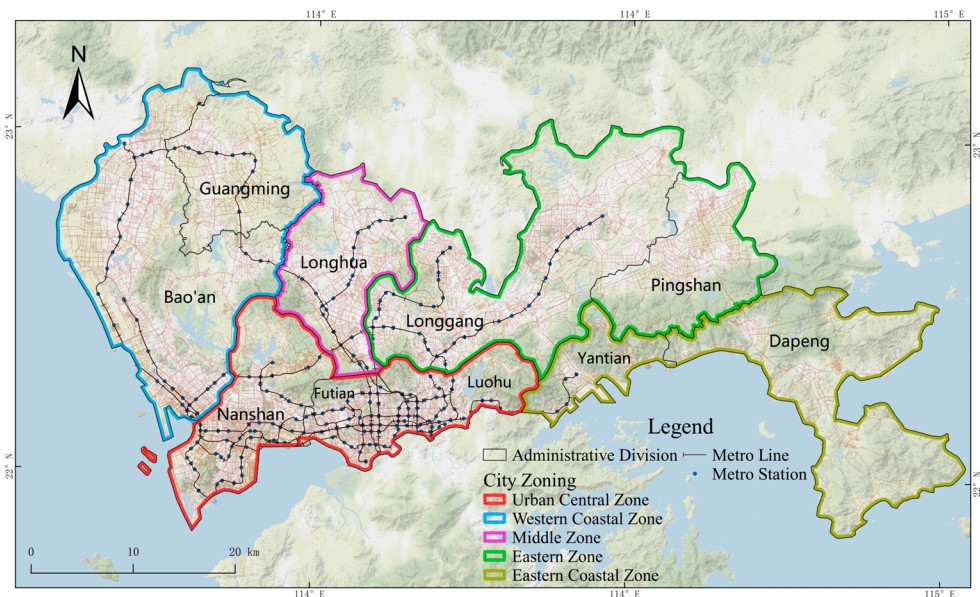

**Figure 1.** Distribution of study area and metro station locations.

### 3.2. Data Sources and Preprocessing

Four main datasets were used in this study: the DBs dataset, the Point Of Interest (POI) dataset, the metro station dataset, and the travel survey dataset.

The DBs data consist of the daily order data of DBs enterprises from the Shenzhen government data liberalization platform (https://opendata.sz.gov.cn/, accessed on 20 April 2022). The data time period is selected from 1 May to 11 May 2021, for a total of about 15.31 million data points, including 5 days of holidays, 5 days of working days, and 1 day of the weekend (Note: Saturday, 8 May 2021, is a workday). The weather during this period was sunny or cloudy, with temperatures between 21–31°C and a wind force of 5–8 on the Beaufort Scale, which is suitable for cycling. There were only scattered additions of asymptomatic patients of COVID-19 in Shenzhen during this period, which had a low impact and can be considered as being able to characterize daily patterns of integration cycling. Each DBs order data point contains the time of unlocking, the time of locking, the latitude and longitude coordinates at the time of unlocking, the latitude and longitude coordinates at the time of locking, the user's unique ID, and the enterprise's unique ID. These data need to be preprocessed with coordinate conversion, data cleaning, and mining before use. This mainly includes (1) coordinate conversion: converting all types of data to the same coordinate system; (2) constructing a metro station parking ring filtering dataset: using the traditional range of DBsMIC orders, in other words, a 100 m range of metro entrances as spare data for the DBs–metro parking domain (metro station domain) screening study; (3) mining data for hidden information: calculate ride time using on/off lock time, use Gaode API for cycling path planning to simulate real cycling paths and obtain cycling distances; (4) delete abnormal values: consider the characteristics of DBsMIC, which includes short to medium distances and short durations. Since some unrealistic short or long-distance movements may be caused by system errors or operator re-assignment of Bikesharing, we comprehensively considered the actual integration cycling time and distance, and relied on expert experience to delete data with travel time less than 1 min or

greater than 30 min, travel distance less than 100 m or greater than 5000 m, and duplicate data [34,35]. After cleaning, a total of about 4.43 million DBs ride tracks were obtained, of which starting cycling and finishing cycling accounted for about half each.

In addition, this paper intends to use POI as travel destinations or origins to correlate with DBs to infer and identify DBsMIC data. The POI dataset was collected from Gaode map, the year of data is 2021, which includes 790,000 points of interest (POI); there are a total of 107,123 points located within walking distance of the metro station domain, and the data are detailed and intensively distributed. Data contain POI name, major class, medium class, minor class, and latitude and longitude coordinate information. Due to the needs of the study, this paper divided the POI data into 11 activity types according to their properties and added the operation time attribute and service power index, as shown in Table 1. The operating hours of each POI type were formulated considering the study by Xia et al. and the actual situation in the study area. For instance, most metro operating hours in Shenzhen are from 6 am to 0 am on weekdays, while the operating hours are extended half an hour later in the evening on rest days; pedestrians usually go for meals at noon and dusk, so restaurant POIs do not appear at other times of the day, etc. [36]. The service power index for each type of POI calculated with reference to Li et al. based on the Shenzhen Area of Interest (AOI) and Tencent User Density (TUD) social media data, reflects the discrepancy in the service capacity of different POI types due to differences in physical area, size, and other attributes [28].

**Table 1.** Mapping between activity types and related POI categories, service power index, and operating schedule for POI categories.

| Code | Type of Activity | Primary POI Categories | The Service Power Index | Workday Relevant Hours | Weekend Relevant Hours |
|------|------------------|------------------------|-------------------------|------------------------|------------------------|
| 1 | Metro Services | Metro entrance | 2 | 6:00–24:00 | 6:00–0:30 |
| 2 | Home | Residential communities | 0.6 | 0:00–24:00 | 0:00–24:00 |
| | | Residential buildings | 0.46 | 0:00–24:00 | 0:00–24:00 |
| | | Hotels | 0.18 | 0:00–24:00 | 0:00–24:00 |
| 3 | Work | Media agencies, insurance companies, finance companies, securities companies, financial and insurance service providers | 0.59 | 8:00–22:00 | 8:00–18:00 |
| | | Large enterprises, general companies | 0.47 | 8:00–22:00 | 8:00–18:00 |
| | | Government, banks, social groups | 0.47 | 8:00–17:00 | Closed |
| | | Industrial Parks | 0.43 | 8:00–22:00 | 8:00–18:00 |
| | | Factory | 0.21 | 8:00–22:00 | 8:00–18:00 |
| 4 | Catering Services | Chinese restaurants, foreign restaurants | 0.43 | 11:00–14:00 17:00–21:00 | 11:00–14:00 17:00–21:00 |
| | | Dessert, cold drink, pastry and other food, and beverage-related establishments | 0.18 | 9:00–22:00 | 9:00–22:00 |
| 5 | Shopping services | Shopping malls, shopping streets, general markets, supermarkets | 0.77 | 9:00–22:00 | 9:00–22:00 |
| | | Building materials market | 0.56 | 9:00–22:00 | 9:00–22:00 |
| | | Electronic shops, flower, bird and fish markets | 0.47 | 9:00–22:00 | 9:00–22:00 |
| | | Supermarkets, convenience stores, clothing, shoes, hats and leather goods stores, personal goods, cosmetics stores, exclusive stores, culture stores, sports stores | 0.3 | 9:00–22:00 | 9:00–22:00 |

**Table 1.** *Cont.*

| Code | Type of Activity | Primary POI Categories | The Service Power Index | Workday Relevant Hours | Weekend Relevant Hours |
|------|------------------|------------------------|-------------------------|------------------------|------------------------|
| 6 | Life services | Telecommunications, electricity, water supply business halls, business halls, post offices | 0.47 | 8:00–17:00 | 8:00–17:00 |
| | | Logistics and express | 0.33 | 9:00–22:00 | 9:00–22:00 |
| | | Baby services, photography printing shops, laundry, travel agencies, beauty salons, car repair sales | 0.23 | 9:00–22:00 | 9:00–22:00 |
| 7 | Science, education, and cultural services | Universities, scientific research institutions, | 1 | 0:00–24:00 | 0:00–24:00 |
| | | libraries | 0.97 | 9:00–22:00 | 9:00–22:00 |
| | | Elementary school, junior high school, high school, kindergarten | 0.43 | 8:00–18:00 | Closed |
| | | Training institutions | 0.29 | 8:00–22:00 | 8:00–18:00 |
| | | Science and education places | 0.24 | 9:00–22:00 | 9:00–22:00 |
| 8 | Sports recreation | Parks, squares | 0.82 | 7:00–22:00 | 7:00–22:00 |
| | | Sports and leisure service venues | 0.81 | 9:00–22:00 | 9:00–22:00 |
| | | Bar, disco, KTV | 0.73 | 14:00–5:00 | 14:00–5:00 |
| | | Tourist attractions | 0.59 | 7:00–22:00 | 7:00–22:00 |
| | | Relaxation areas, bathing and massage facilities | 0.44 | 9:00–22:00 | 9:00–22:00 |
| 9 | Healthcare services | General hospitals, specialized hospitals, emergency centers | 0.65 | 0:00–24:00 | 0:00–24:00 |
| | | Disease prevention institutions, medical and health services, clinics, health and nursing shops | 0.31 | 9:00–22:00 | 9:00–22:00 |
| 10 | Transportation services (excluding subway) | Train stations, coach stations | 0.81 | 0:00–24:00 | 0:00–24:00 |
| | | Bus stops, other transportation-related places, ports, docks | 0.3 | 6:00–23:30 | 6:00–23:30 |
| 11 | Other | Public toilets, public telephones, ATM | 0.18 | 0:00–24:00 | 0:00–24:00 |

The metro line dataset is from Gaode Map and is dated up to June 2021, including 12 metro lines, 234 metro stations, and 1133 metro entrances.

The travel survey dataset is an investigation of residents' DBs travel in the metro station domain of the study area, conducted by a combination of field surveys and online questionnaires. The field surveys use stratified random sampling and select some representative metro station hubs in Shenzhen as offline survey sites. The survey period was from September 2022 to October 2022, with 406 questionnaires returned and 356 valid questionnaires, of which approximately access use and egress use accounted for about half each.

## 4. Research Methods

### 4.1. Overall Idea

DBsMIC can be further divided into access use and egress use. In this paper, the identification method is the same for both access use and egress use, access use is used as an example to illustrate and verify the proposed method for identifying DBsMIC.

This paper proposes a method for identifying the DBsMIC based on an improved E2SFCA and Bayes' rule, and explores the spatiotemporal patterns of DBsMIC. Specifically, the method first screens potential destinations (candidate POIs) for the ride based on the association between the riding parking points and POIs, with constraints of POI operating time and walkable distance. Secondly, the POI service capacity index is introduced to improve the E2SFCA algorithm, which calculates the level of attractiveness of each potential destination to the rider. Then, Bayes' rule is used to calculate the probability of the rider visiting each potential destination. The destination with the highest probability of being visited is identified as the destination of the ride, which allows for the identification of riding

behaviors that are directed toward metro stations. Finally, the spatiotemporal characteristics and distribution patterns of the DBsMIC are analyzed based on the identification results. The specific process is shown in Figure 2.

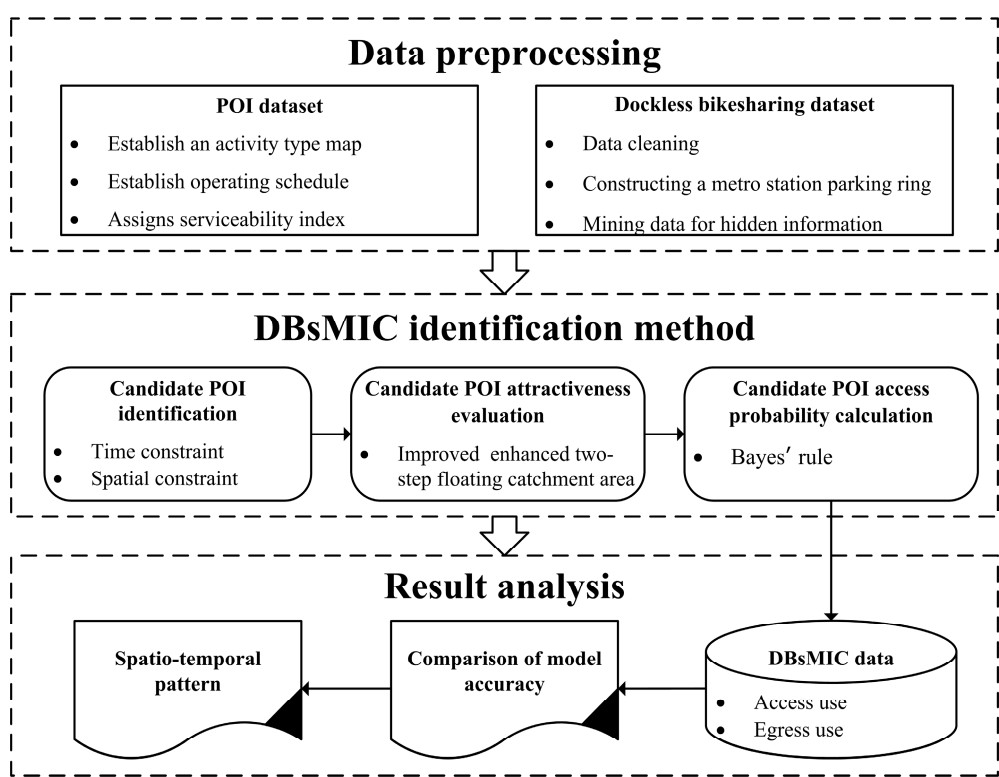

**Figure 2.** Frameworks and workflows.

*4.2. DBsMIC Identification Method*

The basic principle of the DBsMIC identification method designed in this paper is to evaluate the access probability of each candidate destination POI for cycling activities, and use the POI with the highest access probability as the destination of the ride. Bike rides with destinations at metro stations are considered bikesharing–metro integration cycling. The implementation of the method is divided into three steps: candidate POI identification, candidate POI attractiveness evaluation, and candidate POI access probability calculation. The details are broken down as follows.

4.2.1. Candidate POI Identification

Candidate POIs are POIs that are within walkable distance of DBs and in operational status, in other words, potential destinations for DBs riders. As DBs is more flexible and less restricted by road types and parking places, riders can generally park their bikes closer to their destinations with shorter walking distances, so the most comfortable walking distance (CWD) of 200 m is used as the walkable distance [37]. POI operating hours are formulated with reference to relevant literature and combined with the actual situation (Table 1). Figure 3 gives a schematic diagram of how to better understand the candidate POI identification process. Assuming that some DBs is parked in a metro station domain at 11:20 am on a Sunday, POIs that are within walkable distance of their parking points and in operation at that moment will be identified as potential destinations (green) for that ride, as candidate POIs. The formula is expressed as follows.

$$
\begin{aligned}
&if \\
&\quad (d(S, P_{S,i}) \in d_{CWD}) \ and \ (t \in P_{S,i}.h) \\
&then \\
&\quad P_{S,i} \in \ S.CPOIList
\end{aligned}
\tag{1}
$$

where $d(S,P_{s,i})$ denotes the distance between parking point $S$ and POI point $P_{s,i}$, $d_{CWD}$ denotes the most comfortable walking distance, t denotes the lock-off time of parking point $S$, $P_{s,i}.h$ denotes the operating period of POI $P_{s,i}$, and *S.CPOIList* denotes the set of candidate POIs of parking point $S$.

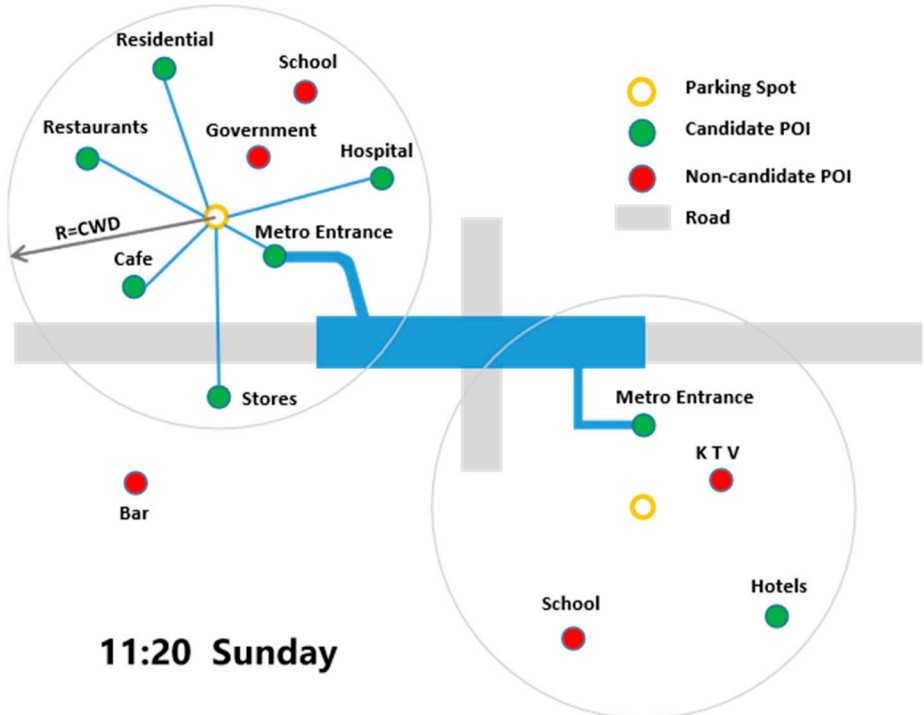

**Figure 3.** Schematic diagram of candidate POI identification.

4.2.2. Candidate POI Attractiveness Evaluation

This paper introduces and improves the E2SFCA to evaluate the attractiveness of candidate POIs to infer DBs trip destinations. Among the various improvements to the traditional 2SFCA, the E2SFCA proposed by Luo et al. is more objective and reliable as it introduces a distance decay function to simulate the impact of accessibility variation with distance within the study area, and it reflects the variation in the accessibility of different areas to the service facilities by setting distance weights in different areas. E2SFCA is not only widely used for accessibility analysis, but also frequently used for inference of taxi travel destinations [26,27,36]. This paper suggests that the method is also applicable for an inference of DBs travel destinations.

In the inference of DBs travel destinations in this paper, supply points of E2SFCA refer to DBs parking points, while the demand points refer to candidate POIs. The attractiveness of non-candidate POIs is set to zero because these POIs are out of reach of bicycle riders in space or time. In order to calculate the attractiveness of candidate POI points for DBs parking spots, this paper improves on the E2SFCA by introducing a POI service power index to compensate for the fact that the original E2SFCA does not consider the differences in size, area, and capacity between various types of POIs. Its calculation steps can be divided into three steps.

1.  Calculate the distance decay weighted supply/demand ratio for parking points. For a parking point $S((x,y),t)$, its walkable distance (CWD) is divided into four sub-regions $d_1$, $d_2$, $d_3$, and $d_4$ with radii of 0–20 m, 20–50 m, 50–100 m, and 100–200 m respectively, and then all candidate POIs within each sub-region are identified and

the weighted supply–demand ratio $R_S$ of the parking point $S$ is calculated with the following equation.

$$R_S = \frac{1}{\sum_{r=1}^{4} \sum_{i \in \{d(S, P_{S,i}) \in d_r\}} f(S, P_{S,i}) W_r} \tag{2}$$

$d_r$ denotes the $r$ sub-region and $W_r$ is the distance weight of the r sub-region, which is calculated here using the Gaussian function proposed by [38], defined as follows.

$$W_r = e^{(\frac{r-1}{4})^2 ln(0.01)}, (r = 1, 2, 3, 4) \tag{3}$$

2. Calculate the candidate POI attractiveness. For the candidate POI $P_{S,i}$, all parking points in different sub-regions within its 200 m range are determined, and the $R_S$ of all parking points in the sub-region are aggregated and multiplied by the distance weight and the service power index $SC_{type}$ of that POI to obtain the self-attractiveness $A_{PS,i}$ of $P_{S,i}$, which is calculated as follows.

$$A_{P_{S,i}} = \sum_{r=1}^{4} \sum_{S \in \{d(S, P_{S,i}) \in d_r\}} R_S W_r SC_{type} \tag{4}$$

3. Calculate the attractiveness of the candidate POI to the parking point. The final candidate POI attractiveness assessment model $G(S, P_{S,i})$ for parking point $S$ is obtained by substituting $A_{PS,i}$ into the basic gravity model with the following equation.

$$G(S, P_{S,i}) = \frac{A_S A_{P_{S,i}}}{d(S, P_{s,i})^\beta} * \alpha, P_{s,i} \in S.CPOIList \tag{5}$$

where $G(S, P_{S,i})$ denotes the correlation between parking point $S$ and candidate POI $P_{S,i}$, which can be used to represent the attraction of $P_{S,i}$ to point $S$; $A_S$ and $AP_{s,i}$ denote the self-attraction of points $S$ and $P_{S,i}$, respectively; $d(S, P_{S,i})^\beta$ (Euclidean distance) denotes the traffic impedance of the two places; $\beta$ is the distance decay coefficient, the larger the value of the coefficient the stronger the correlation between the two points and the distance; $\alpha$ denotes some common influencing factors or normalization parameters.

In this study, $A_s$ is set to 1 for the convenience of subsequent calculations and comparisons. Previous studies have shown that the optimal $\beta$ value is between 1 and 2 [36], and in Zhao et al.'s taxi destination inference, the $\beta$ value is taken as 1.5 [27], while DBs parking points tend to have a closer relationship with distance and more serious distance decay, so the $\beta$ value in this study is taken as 2. However, residents' travel purpose is closely related to time; for example, metro stations stop operating in the early morning, and the probability of residents riding DBs to the metro stations in the early morning is almost zero, so a time function $f(S, P_{S,i})$ is used instead of $\alpha$ to characterize the effect of time on correlation. That is, Equations (5) and $f(S, P_{S,i})$ can be expressed as follows, respectively.

$$G(S, P_{S,i}) = \frac{A_{P_{S,i}}}{d(S, P_{S,i})^2} * f(S, P_{S,i}), P_{S,i} \in S.CPOIList \tag{6}$$

$$f(S, P_{S,i}) \begin{cases} 1 & S.t \in P_{S,i}.h \\ 0 & S.t \notin P_{S,i}.h \end{cases} \tag{7}$$

### 4.2.3. Candidate POI Access Probability Calculation

After obtaining the candidate POI attractiveness based on the above steps, the visit probability of each candidate POI is further calculated using Bayes' rule to infer the purpose

of the DBs trip. Given parking point $S((x,y),t)$, the visit probability of the candidate POI $P_{S,i}$ is as follows.

$$\Pr(P_{S,i} \mid (x,y),t) = \frac{\Pr((x,y) \mid P_{S,i},t) \cdot \Pr(P_{S,i} \mid t) \cdot \Pr(t)}{\Pr((x,y),t)}, P_{S,i} \in S.CPOIList \quad (8)$$

$\Pr(P_{S,i} \mid (x,y), t)$ denotes the probability that if a rider stops at $(x,y)$ at time t then they visit $P_{S,i}$; in general, given $P_{S,i}$, one can assume that the stopping location and time are conditionally independent, so $\Pr((x,y) \mid P_{S,i}, t) = \Pr((x,y) \mid P_{S,i})$, $\Pr((x,y) \mid P_{S,i})$ denotes the probability that a passenger stops at $(x,y)$ if they have $P_{S,i}$ as their destination. $\Pr(P_{S,i} \mid t)$ represents the probability of a resident visit for the type of activity corresponding to $P_{S,i}$ at time $t$. $\Pr((x,y),t)$, represents the probability of stopping at $(x,y)$ at time $t$. $G(S, P_{S,i})$ takes into account time, distance, etc.; therefore, it can be used to estimate $\Pr((x,y) \mid P_{S,i}, t)*\Pr(P_{S,i} \mid t)$ with the following equation.

$$\Pr((x,y) \mid P_i, t) * \Pr(P_{S,i} \mid t) \propto G(S, P_{S,i}) \quad (9)$$

Thus, for a parking point $S((x,y),t)$ and a candidate POI list $S.CPOIList = (P_{S,1}, P_{S,2}, \ldots, P_{S,n})$, the access probability formula for the candidate POI $P_{S,i}$ is

$$\Pr(P_{S,i} \mid (x,y),t) = \frac{G(S, P_{S,i})}{\sum_{i=1}^{n} G(S, P_{S,i})}, P_{S,i} \in S.CPOIL \text{ ist} \quad (10)$$

Finally, based on the above formula, the probability of a rider visiting each candidate POI point after parking at point $S$ can be calculated. The candidate POI with the highest visit probability is selected as the real destination of the trip, and the purpose of the trip is identified according to the mapping relationship between POI and activity type (Table 1), and the DBsMIC data are filtered.

*4.3. Spatiotemporal Characteristic Analysis Method of the DBsMIC*

The analysis of the spatiotemporal characteristics of DBsMIC requires both conventional mathematical and statistical methods to examine the temporal travel patterns of integration cycling, in addition to detecting the travel patterns of integration cycling from a spatial perspective. Considering that the kernel density estimation method is an effective indicator for measuring density differences within a study area [39], this paper introduces this method to study the spatial clustering of DBsMIC. The calculation formula is as follows.

$$F_n(x) = \frac{1}{nh} \sum_{i=1}^{n} k\left(\frac{x - x_i}{h}\right) \quad (11)$$

$$k\left(\frac{x - x_i}{h}\right) = \frac{1}{\sqrt{2\pi}} \exp\left(-\frac{(x - x_i)^2}{2h^2}\right) \quad (12)$$

where $F_n(x)$ is the kernel density value, $h$ denotes the search radius (for an average cycling distance of 1343 m), $n$ denotes the number of analysis unit geometric centers within the search radius, $k$ is the distance weight between analysis unit geometric centers, and $x - x_i$ denotes the distance from the estimated analysis unit geometric center $x$ to the sample analysis unit geometric center $x_i$.

## 5. Results

*5.1. Analysis of the Results of the DBsMIC Recognition*

In order to verify the utility of the method proposed in this paper for identifying DBsMIC, an inference of the purpose of DBs rides by residents in the Shenzhen metro station domain was conducted based on Method 4 and compared with two methods based on improved gravity models by Li et al. (Method 1 and Method 2, which considered the proportion of activity types and POI service capacity based on the gravity model,

respectively) [28], and the E2SFCA method (Method 3) proposed by Ross-Perez et al. [29] The results are shown in Table 2, which shows that among the inferred results of each of the four methods, the proportion of metro services is the highest, far ahead of the second-highest activity type, indicating that all four methods have certain effectiveness in identifying the DBsMIC in the metro station domain. However, there are significant differences in the proportion of metro services among the four methods, so the accuracy of the inferred metro service results of the four methods needs to be further verified. The applicability of the methods based on the gravity model and E2SFCA in the field of inferring the purpose of DBs trips has been demonstrated experimentally by Li et al. and Ross-Perez et al., so the accuracy of the methods in inferring activity types other than metro services will not be discussed further in this paper.

**Table 2.** The purpose of the trip inferred the result.

| Code | Activity Type | Percentage of POI for Activity Type | Method 1 | Method 2 | Method 3 | Method 4 (Method in This Paper) |
|------|---------------|-------------------------------------|----------|----------|----------|---------------------------------|
| 1 | Metro Services | 0.14 | 56.02 | 77.28 | 35.64 | 66.72 |
| 2 | Home | 4.92 | 5.42 | 2.47 | 12.81 | 6.64 |
| 3 | Work | 21.92 | 3.59 | 2.58 | 6.04 | 4.01 |
| 4 | Catering Services | 17.50 | 5.35 | 2.48 | 3.13 | 1.55 |
| 5 | Shopping services | 26.19 | 8.75 | 5.71 | 9.95 | 6.22 |
| 6 | Life services | 19.43 | 6.73 | 2.17 | 8.50 | 3.01 |
| 7 | Science, education, and cultural services | 4.34 | 3.31 | 1.52 | 3.59 | 1.87 |
| 8 | Sports recreation | 2.74 | 2.36 | 2.71 | 2.41 | 2.73 |
| 9 | Healthcare services | 1.07 | 2.42 | 1.56 | 3.78 | 3.17 |
| 10 | Transportation services (excluding subway) | 0.87 | 2.21 | 0.63 | 8.25 | 2.66 |
| 11 | Other | 0.87 | 3.66 | 0.89 | 5.90 | 1.42 |

Accuracy validation is an important means of evaluating the validity of inference results for travel purposes. The four methods of DBsMIC identification results were validated through travel survey data, and the proportion of travel made by the four identification methods and the survey data for different distance intervals of metro service activities in the metro station domain were counted separately, as shown in Figure 4.

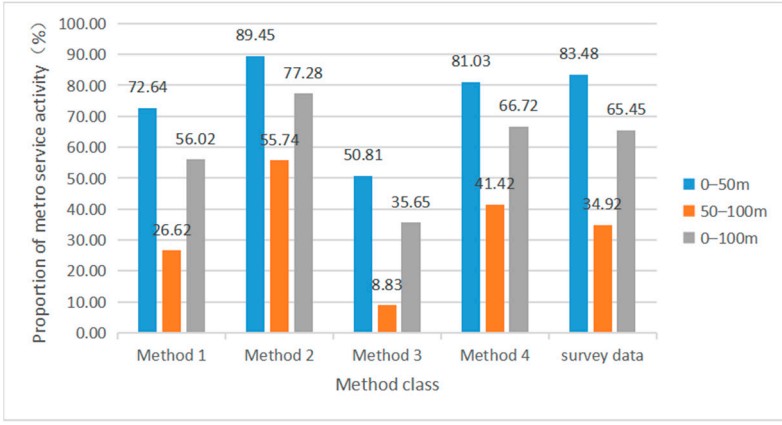

**Figure 4.** Proportion of metro service activity at different distance intervals for the four identification methods and travel survey data.

The identification results show that the results of Method 4 proposed in this paper are more consistent with the real survey data than the other three methods. Method 4 has 81.03% of the DBsMIC within 50 m of the metro entrance, 41.42% for the 50–100-m range, and 66.72% overall. Method 2 had the highest percentage of DBsMIC, with an overall

percentage of 77.28%, while Method 3 had the lowest percentage of DBsMIC, with an overall percentage of 35.65%.

The reasons for this are that Method 1 and Method 3 have a lower percentage of DBsMIC than the real data because they do not consider the impact of POI service capacity. Method 2 and Method 4 both consider POI service capacity, but they have different methods of calculating environmental impact factors. Method 2 expresses environmental factors in terms of activity type density and type ratios, while Method 4 corrects for the ratio of supply and demand of starting and stopping points to POIs' environmental impact factor. The ratio of supply and demand of starting and stopping points to POIs of DBs reflects the aggregation degree of starting and stopping points around different candidate POIs, and the aggregation degree of starting and stopping points around different candidate POIs. This better expresses the attractiveness of each candidate POI compared to the type density and type ratio of POI activities. The raw track data and model identification results also confirm this view; in the raw data, the number of DBs orders within 50 m of the underground entrance is about twice the number of orders within 50–100 m, and the starting and stopping points of DBs are clustered towards the metro entrance. Moreover, from the identification results of the four methods, the proportion of DBsMIC within 50 m of the metro entrance is much larger than the proportion of other activity types and the proportion of DBsMIC outside 50 m. This is consistent with the fact that integration cyclists tend to park and pick up their bikes closer to the metro entrances. However, due to other factors, some integration cyclists still choose to park 50 m away from the metro entrance. For example, some metro entrances are located inside shopping malls or underground plazas, so it is necessary to walk a little further after parking or before picking up a bike. These situations can lead to DBsMIC outside the 50 m range of metro entrances.

### 5.2. Analysis of the Spatiotemporal Characteristics of DBsMIC

Based on the accurate identification of orders for DBsMIC, this paper further analyzes the behavioral and spatiotemporal characteristics of DBsMIC in the study area, to provide a reference for further optimizing the planning and implementation of green mobility policies in the study area.

#### 5.2.1. Cycling Behavioral Characteristics

1.  Overall Cycling Behavioral Characteristics

Statistical analysis of the DBsMIC data identified by Method 4 shows that a total of 2,959,900 DBsMIC trips were generated during the study period. Of these, 1,462,800 were access use and 1,487,100 were egress use, respectively, each accounting for about half of the total number of trips. The average daily number of integration cycling trips was 269,100. Statistics on the number of users who took the integration trips (Table 3) show that a total of 976,996 users completed DBsMIC trips during the study period. The number of user integration trips follows an exponential distribution. However, in absolute terms, there are still 49,000 users who have ridden 10 times or more and have a stable user base.

**Table 3.** User cycling.

| Number of Rides | Number of Users | Percentage(%) | Number of Rides | Number of Users | Percentage (%) |
|---|---|---|---|---|---|
| 1 | 446,376 | 45.69 | 6 | 34,728 | 3.56 |
| 2 | 179,345 | 18.36 | 7 | 25,737 | 2.63 |
| 3 | 94,188 | 9.64 | 8 | 19,749 | 2.02 |
| 4 | 65,763 | 6.73 | 9 | 14,678 | 1.50 |
| 5 | 47,341 | 4.85 | ≥10 | 49,091 | 5.02 |
| | Total | | | 976,996 | |

To better understand the dynamic characteristics of the DBsMIC, we conducted kernel density analysis and time variation analysis separately for distance and time, as illustrated

in Figure 5. The analysis reveals that the average cycling time is the shortest during the morning peak, and the average integration cycling time on weekdays is shorter than on weekends and holidays. Additionally, the integration cycling time is the longest on holidays. Furthermore, cycling times in the afternoon and evening are longer than in the morning.

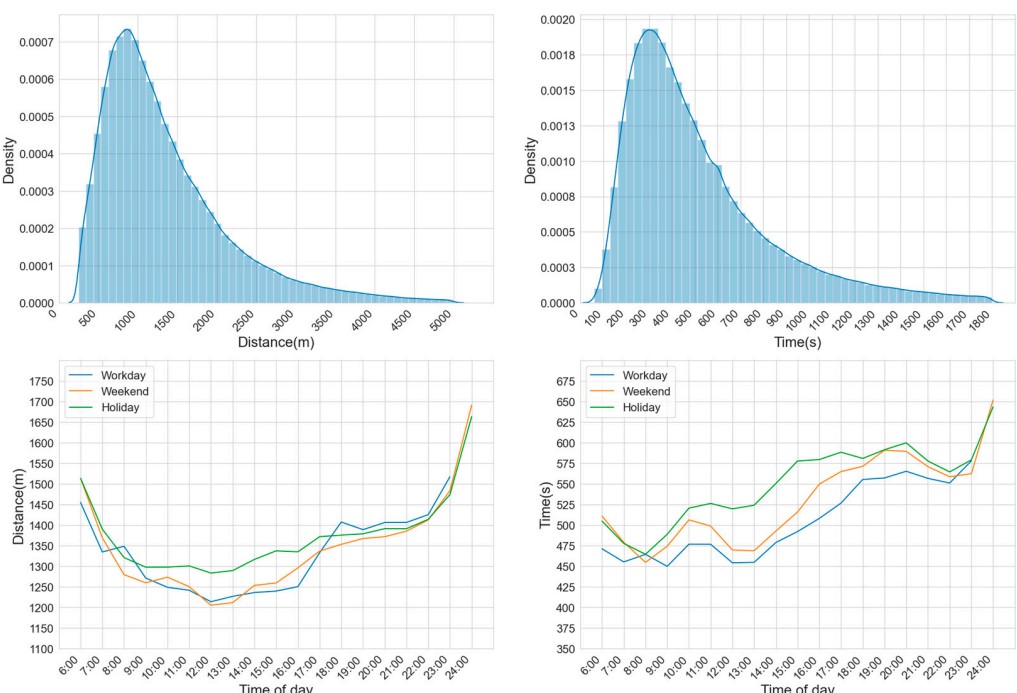

**Figure 5.** Distribution of cycling distance, cycling time kernel density, and time variation for DBsMIC.

2. Riding Volume Time Characteristics

Figure 6 shows the distribution of daily DBsMIC and their average cycling per hour interval during the study period. The number of integration cycling trips on rest days is significantly lower than that on weekdays, with the first and last days of the May Day holiday being at the peak of leisure and tourism travel, and the demand for metro integration cycling substantially lower in the middle of the holiday. The change in the distribution of integration cycling throughout the day reflects the morning and evening peaks of metro integration cycling, with the morning peak lasting shorter between 7:00 and 9:00 and the evening peak lasting longer between 17:00 and 20:00, which is due to the concentration of work hours in the morning and the irregular hours of some employees at the end of the day, making the morning peak significantly higher than the evening peak. The evening peak is more evenly distributed, with the evening peak falling more gently.

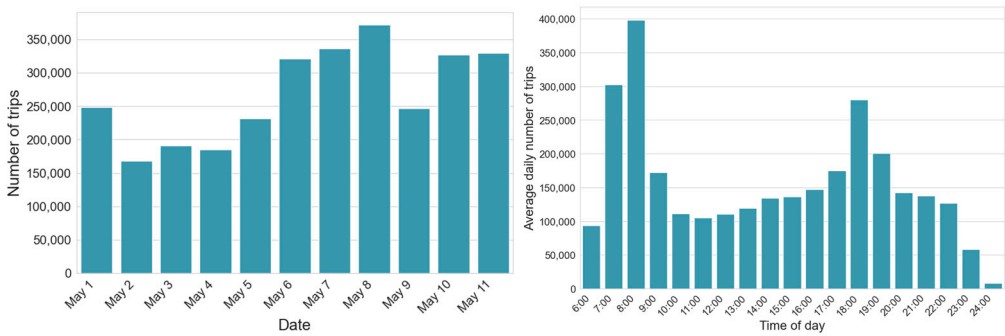

**Figure 6.** Distribution of daily cycling and hourly average cycling for DBsMIC.

Comparing the distribution of the average daily ridership of DBsMIC on weekdays, weekends, and holidays in terms of hourly segments as shown in Figure 7, it can be seen that weekdays, weekends, and holidays all show different degrees of morning and evening peak characteristics, and the morning peak ridership of DBsMIC is greater than the evening peak, but the change in morning and evening peak ridership of DBsMIC on weekdays is greater than that on weekends and holidays, and the ridership of DBsMIC on weekdays in both periods accounts for 55% of the whole day ridership of DBsMIC. This indicates that DBsMIC plays a significant role in commuting on weekdays, alleviating motor vehicle congestion to a certain extent during the morning and evening peak periods. Whereas for weekends and holidays, although the same morning and evening peak characteristics exist, the change in ridership is more moderate and the overall change throughout the day is not significant. This is because, unlike weekday commuting, non-weekday travel is more for leisure and recreation, and is less constrained by time, so ridership is more evenly distributed. In addition, the change in ridership between weekend and holiday hours is basically the same, with weekend ridership slightly greater than holiday ridership overall, so weekends and holidays will be combined into rest days in subsequent analysis.

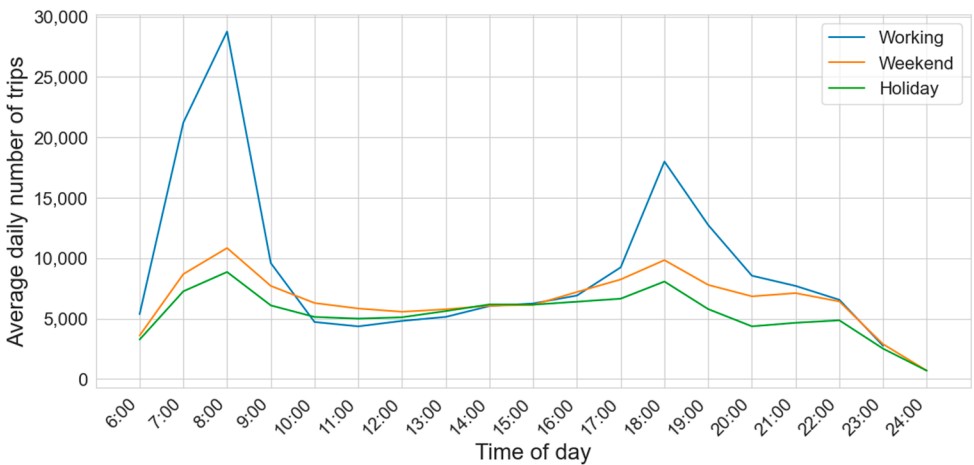

**Figure 7.** Average daily ridership of DBsMIC by hour.

Figure 8 analyzes the hourly distribution of access use and egress use for DBsMIC on both weekdays and rest days. The analysis shows that access use and egress use on weekdays and rest days conform to the morning and evening peak characteristics, However, the morning peak for access use occurs one hour earlier than the morning peak for egress use, which is also in line with the pattern of residents using the metro for commuting.

5.2.2. Analysis of the Spatiotemporal Characteristics of Cycling

1.    Overall Spatiotemporal Cycling Characteristics

The DBsMIC full sample refers to the total number of integration cycling trips in and out of metro stations during the study period. It was analyzed for kernel density and counted for each metro station, and its spatial distribution is shown in Figure 9. The figure indicates a clear aggregation phenomenon of DBsMIC in several business districts and employment centers, such as Baoan Centre, Xili University City, Nanshan-Houhai Hi-Tech Park, among others. Several stations with high demand for integration cycling are identified, such as Xixiang, Bihaiwan, Gushu, Baoan, and Baoan Centre in Baoan District; Nanshan, Houhai, Xili, and Liuxian Dong in Nanshan District; Hongshan and Longsheng in Longhua District; Yuanfen in Longgang District; and Wuhe in Longgang District, among others. In contrast, some metro stations in the suburbs have low demand for integration cycling due to being recently built and located at the edge of the DBs operation area, resulting in incomplete supporting facilities. These stations are mainly found in the northern half of Metro Line 3, Line 4, and Line 10.

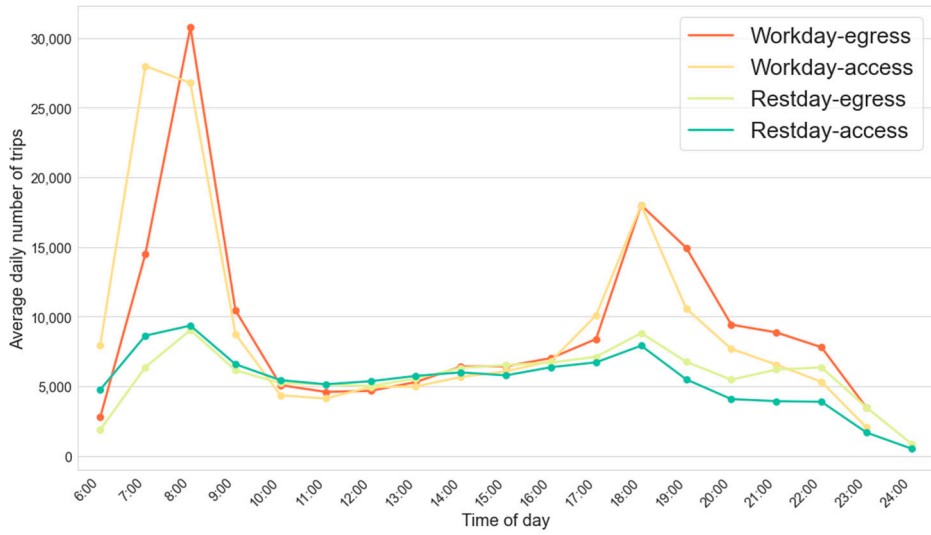

**Figure 8.** The hourly distribution of daily average ridership for access use and egress use of DBsMIC.

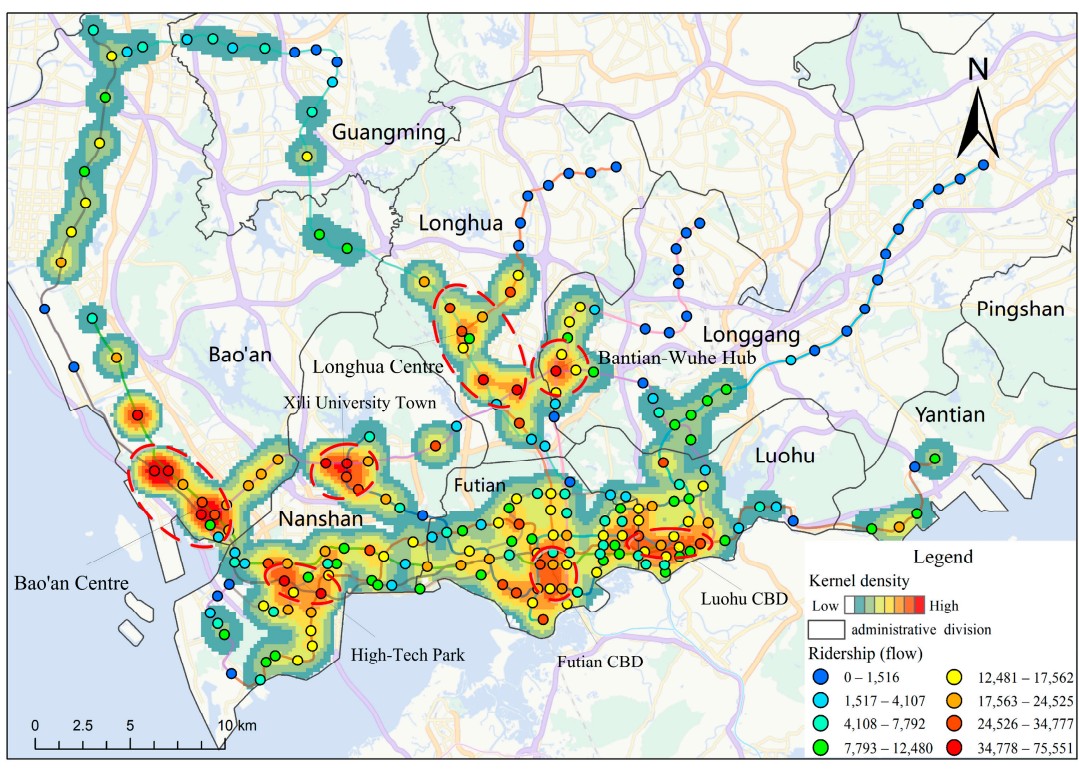

**Figure 9.** Statistics and kernel density distribution of DBsMIC's full sample ridership.

2.    Spatiotemporal Characteristics of Cycling on Working Days and Rest Days

The study period was divided into two parts: weekdays and rest days. The average daily ridership of DBsMIC at each metro station was calculated, and kernel density analysis was conducted to examine the spatial distribution. The results are shown in Figure 10. It is evident that the overall integration cycling on rest days decreased significantly compared to weekdays. However, the main gathering areas remained unchanged and were concentrated in several business districts. Notably, some individual metro stations, such as the Youth Palace, Shatoujiao, and Shenzhen Bay Park, showed relatively higher integration cycling on rest days. This might be due to the fact that residents tend to travel for leisure and entertainment on rest days, and these metro stations are surrounded by abundant leisure and entertainment facilities with strong attractions.

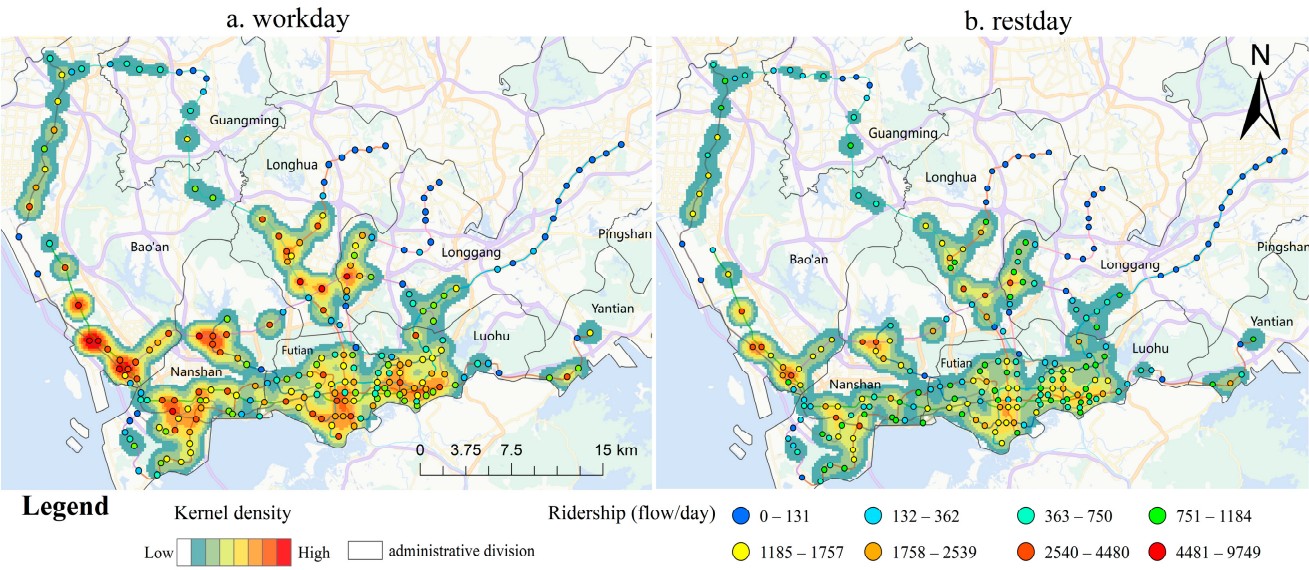

**Figure 10.** Kernel density distribution and statistics of average daily ridership for DBsMIC by metro station on weekdays and rest days.

3.  Spatiotemporal Characteristics of Cycling in and out of the Station during the Morning and Evening Peaks on Weekdays

By counting the hourly access use and egress use ridership at each metro station during the morning and evening peaks on weekdays and conducting kernel density analysis, we observed a clear tidal wave phenomenon, as shown in Figure 11. During the morning peak hours, a large number of residents in the southern part of Longhua District, the western part of Longgang District, and the southern part of Baoan District use DBs rides to enter the metro stations, with egress use concentrated in Nanshan High-Tech Park, Xili University City, and Baoan Centre, where the access use demand is greater than the egress use demand. In the evening peak, the frequency of access use and egress use is the opposite of the morning peak. The hourly volume of access use and egress use during the evening peak hours is nearly half of that during the morning peak hours. This is because residents return from the city center to their suburban residences after work, and the demand for egress use is greater than the demand for access use. Combined with Figure 12, the statistical distribution of the difference between access use and the egress use per hour in the morning and evening peaks (access use minus egress use, warm shades represent access use over egress use, cool shades represent egress use over access use), can demonstrate a complete weekday DBsMIC commuting chain. During the morning peak hours, residents in the suburbs, where rents are relatively cheap and metro stations are more sparse, use DBs to ride to metro stations, then take the metro to the central city metro stations and walk to work; during the evening peak hours, the demand for access use decreases, and residents tend to walk from their workplace to the metro station and take the metro to their place of residence before riding DBs home.

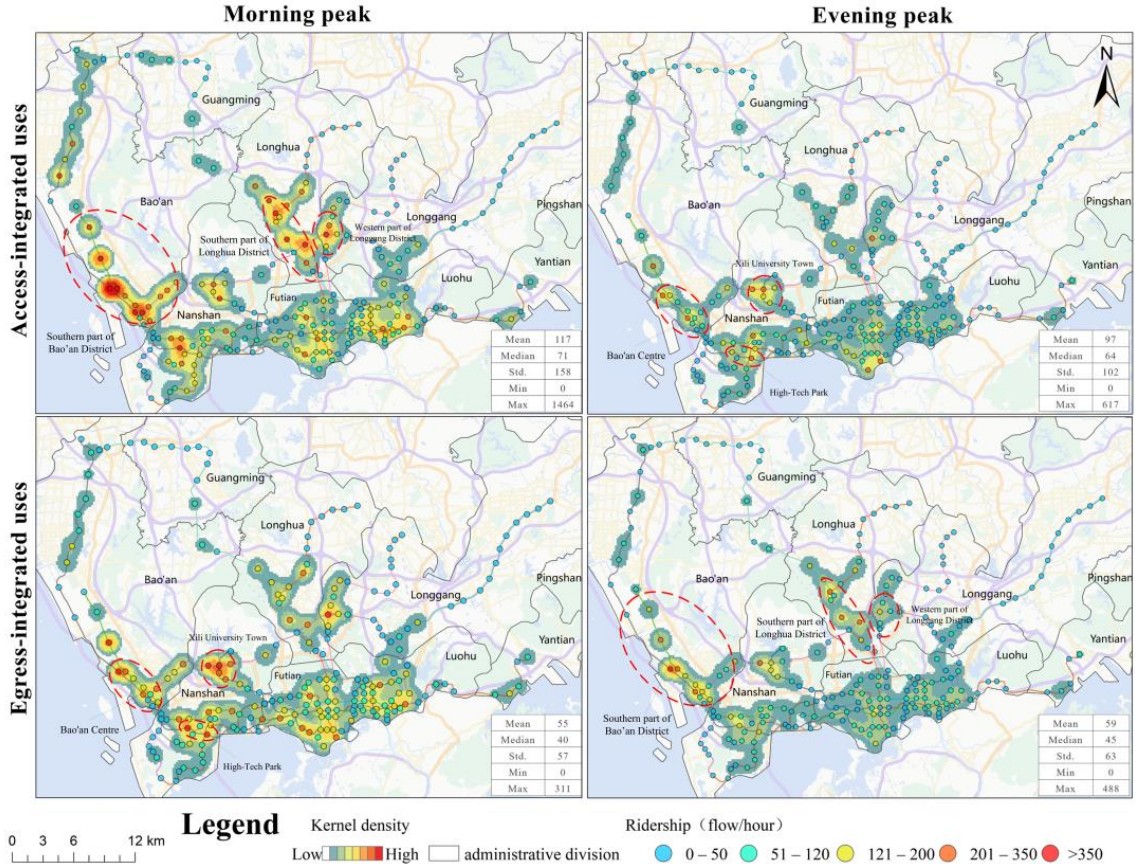

**Figure 11.** Morning and evening peak access use and egress use ridership statistics and kernel density distribution by metro station on weekdays.

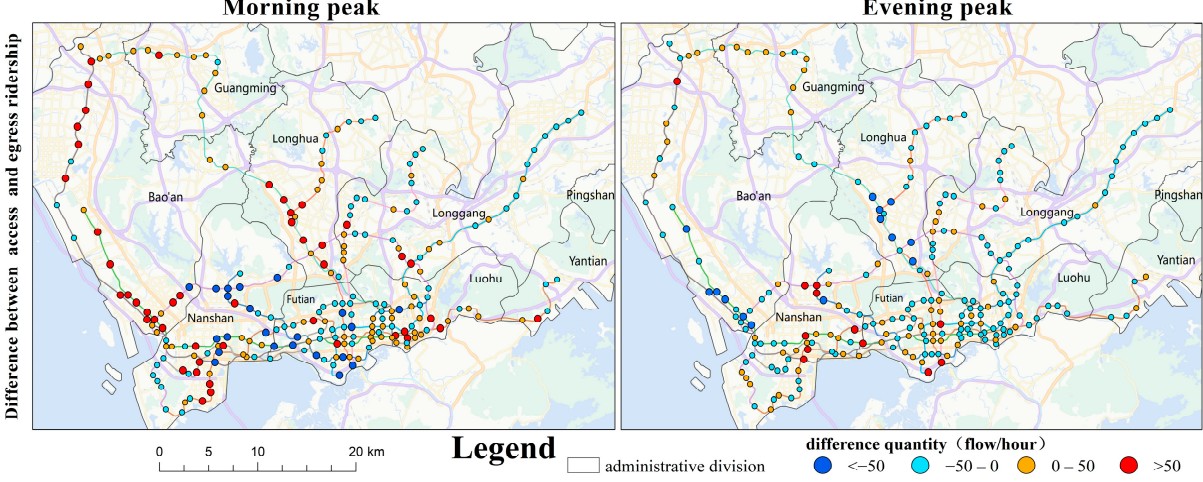

**Figure 12.** Spatial distribution of the difference in the amount of morning and evening peak integration cycling access use and egress use of each metro station on weekdays.

## 6. Discussion and Conclusions

### 6.1. Discussion

The contribution of this study is to propose a method for identifying DBsMIC based on an improved E2SFCA and Bayes' rule, and explores the spatiotemporal patterns of residents' DBsMIC in Shenzhen. The proposed method achieves higher accuracy compared to those proposed by Li and Ross-Perez [28,29]. Furthermore, this study finds that the definition of "orders that start or end within 100 m of the entrance of the metro station as "DBsMIC"

used in previous studies (Li et al., Wu et al., Guo et al., and OFO company [6,23,25]) includes nearly one-third non-DBsMIC orders, resulting in a large error, Although Chen et al. narrowed this range to 50 m, it does not fully fit the actual situation, as it ignores integration cycling orders beyond 50 m [13]. In contrast, the proposed method in this study filters out the non-DBsMIC orders within 100 m, resulting in a more accurate reflection of the actual travel patterns in daily life.

In addition, the results of the spatiotemporal analysis show that there is a tidal phenomenon in the DBsMIC in Shenzhen, with significant heterogeneity in the volume of rides for DBsMIC between the morning and evening commuting peaks, as well as between the city center and the suburbs. This can be attributed to factors such as the distribution of jobs and housing, population density, the density of metro stations, the condition of the road network, and property prices, which are closely related to the built-up environment and social economy of the city. Shenzhen is a pan-central city, with three central urban areas (Nanshan, Futian, and Luohu) that are home to many commercial enterprises and densely populated with metro stations, but also have high property prices and rents due to economic growth. This has resulted in a tidal wave of urban commuting as residents live and work in different areas. The metro stations in the suburbs are sparsely located and commuters have longer transfer distances, which are suitable for DBs. In contrast, the city center is more densely covered by metro stations with shorter transfer distances, making it more suitable for walking. Consequently, the demand for access use is greater than the demand for egress use in the morning peak, and the demand for egress use is greater than the demand for access use in the evening peak. The lower demand for integration cycling in the evening peak can be attributed to two possible reasons. Firstly, because of irregular timing of leaving work there is a more even flow of people. Secondly, because the human body is more energetic in the morning they may be more inclined to ride the connection, taking into account time constraints. Conversely, in the evening residents are physically tired after a day's work and are more likely to choose leisurely walking when they have plenty of time.

Based on these findings, it is recommended that DBs companies consider redeploying their bicycles in the early hours of the morning. For suburban metro stations with high residential density, such as Gushu, Longhua, and Bantianbei, the redundantly parked bicycles in the metro station area in the early hours of the morning should be evenly dispersed to the entrance of the surrounding 2000 m residential area. Only a small portion should be left behind to prevent residents from being unable to find bicycles during morning peak hours and to avoid congestion caused by a buildup of bicycles at the metro entrances. Furthermore, transportation regulators should increase the length of bicycle paths within the 2000 m area around the metro station to encourage residents to use DBsMIC. It is also important to plan parking locations for DBs at the metro entrance to prevent congestion caused by disorderly parking during morning and evening peak hours.

Limited by data acquisition, this study also has certain shortcomings: firstly, regarding the identification method of DBsMIC, this paper does not combine it with traditional survey data, while the accuracy of identification can be improved by modifying the model with a large sample of survey data [40]. Secondly, the experimental data in this paper only adopt the data of DBs orders for 11 consecutive days in Shenzhen, without exploring the patterns in different months and seasons. In addition, this paper only analyzed the spatial and temporal distribution patterns of residents' DBsMIC trips and travel patterns but did not further investigate the influence mechanism. Therefore, subsequent research will focus on further improving the accuracy of the identification method, using a larger sample size for empirical analysis, and further exploring the patterns and influences of residents' DBsMIC.

### 6.2. Conclusions

Exploring the features of DBsMIC helps to gain a deeper understanding of residents' travel patterns, which in turn helps managers improve the effective redistribution of bicycles, promote the coupling efficiency between transportation modes, and achieve

sustainable development of urban transportation. This study proposes a method for identifying DBsMIC based on an improved E2SFCA and Bayes' rule and to compares it with three other identification methods to detect the effects of different factors on DBsMIC identification, and to validate the higher accuracy of the proposed method using data from a survey of residents' DBs trips. The results show that simply representing the DBsMIC ridership by the number of DBs ridership orders within 50 m or 100 m of the metro entrance is inaccurate and has large errors.

Based on the DBsMIC data obtained from the study area, an exploration of the spatiotemporal patterns of DBsMIC of Shenzhen residents revealed the following: (1) In terms of timing characteristics, the morning and evening peaks on weekdays are significant, while the rest days are more moderate. (2) Spatial distribution of ridership reflects the imbalance between urban and suburban areas of Shenzhen and the morning and evening peak ridership of different stations.

**Author Contributions:** Hao Wu established the unique idea and methodology for the current study; Yanhui Wang contributed guidance on paper presentation and logical organization; Yuqing Sun assisted in the translation and grammar check of the thesis; Duoduo Yin, Zhanxing Li, and Xiaoyue Luo provided review and editing of manuscript articles. All authors have read and agreed to the published version of the manuscript.

**Funding:** National Natural Science Foundation of China (42171224,41771157), National Key R&D Program of China (2018YFB0505400), the Great Wall Scholars Program (CIT&TCD20190328), Key Research Projects of National Statistical Science of China (2021LZ23), Young Yanjing Scholar Project of Capital Normal University, and Academy for Multidisciplinary Studies, Capital Normal University.

**Institutional Review Board Statement:** Not applicable.

**Informed Consent Statement:** Not applicable.

**Data Availability Statement:** Not applicable.

**Acknowledgments:** The author appreciates the editors and reviewers for their comments, suggestions, and valuable time and effort in reviewing this manuscript.

**Conflicts of Interest:** The authors declare no conflict of interest.

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
