# Peer review of "Identification and Spatiotemporal Analysis of Bikesharing-Metro Integration Cycling"

_ijgi, doi:10.3390/ijgi12040166_

Round 1

Reviewer 1 Report

Dear authors,

thanks for the submitting your paper. Your research sounds interesting but, prior publication, I suggest you to make some major improvements on the manuscript, namely:

1. It's not clear to the reader which problem you want to address. You state that "accurate identification of the DBS and Metro connection behaviour" is fundamental and that it can help to "further optimising the planning and implementation of green mobility policies". But why is that? What is the problem? Are DBS not used in the correct way? Is the metro system missing customers? Or else? You perform a very complicated and interesting study, but why do you do that? What is the link between the outcome of your analysis and the optimisation of planning and implementation of green mobility policies? Or are you simply explaining spatial pattern based on DBS data?

2. It seems you have made two separate and quite different types of analysis. The first applying/proposing a E2SFCA model, the second (5.2) simply based on descriptives of the DBS data. It's not very clear what is the relationship between these two. Does the second reinforce the results of the first one? Or ohterway around? It looks like you can write two different articles.

Some minor issues:

- most of your literature is based on Chinese examples; the paper would benefit from a more international literature;

- please, define very well what you mean for "urban public bicycles". Each country might have a different definition of that.

- I understand Shenzhen is a very large city, but including 790,000 POI sounds like including everything is on the map....

- In section 4.1 it is very unclear your description of the method behind the E2SFCA. It would help if you rewrite your description using the terms access and egress like in the international literature.

- Figure 2 is very complicated, please try to simplify it.

- 5.1: please say something more about that survey (line 400-401). Now the reader has no information about it, just a figure with some results.

Reviewer 2 Report

They should review the format of the references in the first sections. Remember the journal template. Also the references at the end of the article. 

I think it is important that you introduce a discussion comparing your research with other recent research and separate it from the conclusions. Use the conclusions to resolve the hypotheses you state in the introduction.

The bibliography you use is very extensive and I think it is correct.

Use less or eliminate personal pronouns such as "we". A more impersonal style is better.

Reviewer 3 Report

This paper proposes a DBS and the Metro connection travel identification method based on an improved enhanced two-step floating catchment area (E2SFCA) and Bayes rule, and analyzes the spatial and temporal characteristics of the bike-sharing and the Metro connection behavior in Shenzhen according to the identification results. The following points can be improved.

1. The title is “Identification Method and Spatiotemporal Characteristic Analysis on Dockless Bike-sharing and Metro Connection Behavior: A case study in Shenzhen”. The title is too long and the study object “Dockless Bike-sharing and the Metro Connection Behavior” should be described in a more simple and shorter manner.

 2. Section 4.2 is much longer than section 4.3.

3. How to determine the thresholds in the deletion abnormal values described in lines 200-205?

 4. In lines 472-473, " The average cycling time in the morning peak is 472 the longest. The average cycling time in the morning peak is the shortest, the overall…" both “The average cycling time”

 5. lines 518-519: the morning peak time for entrance connection cycling is one hour earlier than that for the exit. Figure 8: the red line is "workday-START" and the yellow line is “workday-END”, the yellow line is earlier than the red line, please check it.

 6. Figure 10 should add the sub-titles, for example, (a) weekday, (b) rest day

 7. "Nanshan Hi-Tech Park, Xili Liuxian Dong and Baoan Centre" mentioned in line 568 does not appear in Figure 11.

Round 2

Reviewer 1 Report

Dear authors,

thanks for your revised version. You have addressed all my suggestions.

Author Response

Thank you for your affirmation. Your previous comments have greatly improved our paper.